# Plant Molecular Responses to Potato Virus Y: A Continuum of Outcomes from Sensitivity and Tolerance to Resistance

**DOI:** 10.3390/v12020217

**Published:** 2020-02-15

**Authors:** Špela Baebler, Anna Coll, Kristina Gruden

**Affiliations:** National Institute of Biology, Večna pot 111, 1000 Ljubljana, Slovenia; anna.coll@nib.si (A.C.); kristina.gruden@nib.si (K.G.)

**Keywords:** *Potato virus Y*, *Potyviridae*, potato, *Solanum tuberosum*, Solanaceae, plant immune signaling, plant hormones, tolerance, susceptibility, resistance

## Abstract

Potato virus Y (PVY) is the most economically important virus affecting potato production. PVY manipulates the plant cell machinery in order to successfully complete the infecting cycle. On the other side, the plant activates a sophisticated multilayer immune defense response to combat viral infection. The balance between these mechanisms, depending on the plant genotype and environment, results in a specific outcome that can be resistance, sensitivity, or tolerance. In this review, we summarize and compare the current knowledge on molecular events, leading to different phenotypic outcomes in response to PVY and try to link them with the known molecular mechanisms.

## 1. Introduction

Potato virus Y (PVY) is the most economically important virus affecting potato production worldwide [1]. It severely affects potato production in terms of crop yield and quality, which, in the case of secondary infections, can reach yield reductions up to 85% [2,3]. The virus is also infecting other agronomically important crops from the Solanaceae family such as tobacco, pepper, and tomato. The fact that PVY is transmitted by 65 different aphid species in a nonpersistent manner makes the control and prevention an ongoing challenge [4].

PVY, a member of the genus *Potyvirus*, family *Potyviridae* is a flexuous rod-shaped virus with a 9.7 kb positive-sense single-stranded RNA (ssRNA) genome which contains two open reading frames (ORF) encoding 11 functional proteins. One large ORF encodes a polyprotein that is cleaved by virus-specific proteases into 10 functional proteins. In addition, the protein P3N-PIPO is produced by a short overlapping coding sequence [5]. The virions have the RNA genome encapsidated by multiple copies of the coat protein (CP) and covalently linked at the 5′ end to the viral protein genome-linked (VPg). The high-resolution cryoelectron microscopy structure of the PVY virions was recently determined and it showed a left-handed helical arrangement of CPs assembled around viral ssRNA. The structure of CP revealed its intrinsic plasticity, which is most pronounced within extended terminal regions [6]. This plasticity might explain the multifunctional nature of the CP. In fact, this viral structural protein is involved in several steps of virus infection, including assembly, replication and translation, cell-to-cell movement, and long-distance transport [7,8,9]. Similarly, other viral proteins are multifunctional and involved in several stages of the viral cycle by establishing dynamic interactions with viral proteins, RNA and host proteins [10,11].

The complex interaction between plants and viruses includes the interplay of several mechanisms: (1) Viral hijacking of host factors required for efficient infection, (2) mechanisms of plant defense, and (3) mechanisms of viral counterdefense, to circumvent plant defense. The balance between these mechanisms defines a specific outcome that can be beneficial for the virus (compatible interaction) or for the plant (incompatible interaction). In the case of compatible interaction, the virus can multiply and spread within the plant; in susceptible plants, local and systemic symptoms appear, while tolerant genotypes develop no or very mild symptoms (i.e., earlier senescence of infected leaves [12]). In an incompatible interaction, the defense response restricts the viral multiplication and/or spread. There are three forms of resistance against PVY: Extreme resistance (ER), susceptibility genes (S-genes)-conferred resistance, with no visible symptoms or very limited necrosis, and hypersensitive response (HR)-conferred resistance, which is manifested by the formation of local necrotic lesions [13]. In potato, ER, HR, tolerant, and susceptibility responses are described (Figure 1) but no example of natural S-gene mediated resistance is currently known.

The outcome of the interaction depends on the potato genotype, the environment, and the viral strain. Potato cultivars have different genetic backgrounds, resulting in different responses to the virus. It was shown that a single mutation can shift the outcome [14]. On the other hand, studies of PVY strains showed an exceptional diversification via nucleotide mutation and genome recombination resulting in new strains and isolates with different degrees of pathogenicity [1,15,16,17]. Often, outcomes of the interaction in the same potato genotype are different, depending on PVY strain, as was shown for PVY^N^ and PVY^NTN^ in cvs. Igor and Nadine [18] and PVY^N-Wilga^ and PVY^NTN^ in cv. Etola [19]. On the other hand, the appearance of symptoms did not differ in HR response to PVY^N605^-GFP, PVY^NTN^_,_ and PVY^N-Wilga^ in cv. Rywal [20]. There are also many examples that the outcome of the interaction is affected by abiotic environmental factors [21] and in mixed infections with other viruses ([22,23,24], see also review [25]).

Understanding of molecular mechanisms underlying those outcomes is of utmost importance for resistance breeding without growth trade-offs and adaptation of agronomical practices. The studies in the model plants uncovered several aspects of molecular mechanisms in plant immunity [26]. Although some findings can be transferred to crop species using orthology [27,28], this is not always the case as, for example, in recently reported redundancy of Phytoalexin-deficient 4 (PAD4) in Solanaceae [29]. Therefore, it is important to perform the studies also in crop plants, such as potato. Studies of immune responses in potato, especially in combination with PVY are scarce. One of the reasons lies in the complexity of the potato genome, the cultivated cultivars being highly heterozygous autotetraploids [30]. So far, the genomes of double monoploid clone from *Solanum tuberosum* ssp. *Phureja* and diploid homozygous genotype Solyntus from ssp. *tuberosum* were sequenced (The Potato Genome Sequencing Consortium, 2011, https://www.plantbreeding.wur.nl/Solyntus/). In addition, the pan-transcriptome constructed from several potato cultivars is available [31]. 

In the following sections, we compared the molecular responses of plants to PVY and linked them with the known molecular mechanisms of plant immunity (Figure 2).

## 2. Molecular Mechanisms Underlying Resistance to PVY

Plants depend on a sophisticated multilayer immune system to combat virus infection. Innate immunity, RNA silencing, translational repression, and ubiquitination-mediated and autophagy-mediated protein degradation are the major defense mechanisms against viruses in plants [32]. The RNA silencing pathway is the major, evolutionarily conserved antiviral mechanism in plants. As a counter-defense, viruses have adapted by encoding silencing-suppressor proteins that suppress the silencing-based antiviral mechanism (reviewed by [33]), rendering them successful pathogens. Recently, it has been accepted that the first layer of the innate immunity against pathogenic bacteria that relies on the recognition of microbial- or pathogen-associated molecular patterns (MAPMs or PAMPs) by transmembrane pattern recognition receptors (PRRs), PAMP-triggered immunity (PTI) [34], also acts against viruses [35,36,37]. However, how viral double-stranded RNA (dsRNA) could be sensed during PTI is still unknown [38]. As a second layer of the immune system, plants also employ effector-triggered immunity (ETI) against viruses [33]. It involves the specific recognition of pathogen-derived molecules (effectors) by intracellular receptors, known as resistance proteins (R) that results in disease resistance [39]. In potato, dominant R genes can provide two main types of resistance against PVY: ER is conferred by *Ry* genes and HR is conferred by *Ny* genes [40]. Upon recognition, signaling events similar to PTI and ETI responses to other pathogens are triggered. Changes in oxidative burst and Ca^2+^ flux were detected and downstream mitogen-activated protein kinase (MAPK) cascade is activated, resulting in hormonal changes, such as increased level of salicylic acid (SA) and transcriptional reprograming, ultimately leading to pathogen arrest (reviewed by [33]). There were also some additional components participating in the immune response against potyviruses identified. The two endoplasmic reticulum stress pathways, IRE1/bZIP60 pathway and Bax inhibitor 1 (BI-1) pathway, suppress accumulation of potyviruses Arabidopsis and *Nicotiana benthamiana* plants [41].

While mechanisms of pathogen arrest are well understood in Arabidopsis response to some bacteria [42,43], fungi, and oomycetes [44], these mechanisms were described only for some viral pathosystems. For example, it was shown that constitutive, but not pathogen inducible, β-1,3-glucanase-controlled callose deposition at plasmodesmata is blocking the spread of the virus in Arabidopsis interaction with tobamoviruses [45].

Another resistance mechanism was found for Arabidopsis–begomovirus interaction, where translational repression blocks the spread of the virus. Activation of leucine-rich repeat receptor-like protein kinase (LRR-RLK) NIK1 leads to global translation suppression and translocation of the downstream component RIBOSOMAL PROTEIN 10 to the nucleus, where it interacts with a newly identified MYB-like protein, leading to the shutdown of global cellular and viral protein synthesis [46]. In the interaction of plants with potyviruses, a different type of translational repression was detected. *Beclin1*, one of the core autophagy-related genes that were upregulated by viral infection, interacts with nuclear inclusion b (NIb), the RNA-dependent RNA polymerase of turnip mosaic virus, and restricts viral infection through suppression of the viral NIb [47] resulting in less effective translation of viral proteins.

Several atypical dominant resistance genes were identified in the interaction of plants with potyviruses. For example, atypical thioredoxin (with modified active site residues) suppresses sugarcane mosaic virus RNA accumulation [48]. This resistance is not dependent on SA or jasmonic acid (JA) signalling pathways. Another type of R-gene was identified in soybean and confers broad-spectrum resistance to potyviruses. The *Rsv4* gene encodes an RNase H protein that is able to interact with viral RNA polymerase complex and, thus, most probably degrades viral RNA when in dsRNA form [49].

Resistance can also be conferred by S-genes. These are the genes the lack of which renders plants resistant. Their function is either to support viral infection cycle or to be negative regulators of immunity in plants. In contrast to R-gene mediated resistance, this type of resistance is recessive. The most studied S-gene is the alternative isoform of the eukaryotic translation initiation factor 4E (eIF4E), required for potyviral translation initiation, but dispensable for plant growth [50].

### 2.1. Hypersensitive Response-Mediated Resistance: Response at the Right Place at the Right Time

Many different genes for HR-mediated resistance to PVY have been introduced to potato from wild relatives, *Ny, Nc*, and *Nz*, conferring resistance to PVY^O^ (ordinary strains), PVY^C^ (C strains), and PVY^Z^ (Z strains), respectively (reviewed by [13]). New variants of resistance genes have been discovered lately, harboring resistance also towards more aggressive recombinant viral strains, such as *Ny-1* in cv. Rywal [51], *Ny-2* in cv. Romula [52], and *Ny-Smira* in cv. Sárpo Mira [53]. None was, however, characterized on the sequence level.

Localized programmed cell death (PCD) that leads to the appearance of macroscopically visible localized tissue necrosis (Figure 1) is the feature distinguishing HR resistance from ER [54]. Although HR-associated PCD was shown to restrict pathogen spread in some biotrophic pathosystems, it is not required for resistance in several viral pathosystems (reviewed by [54]). This was also shown in potato (cv. Rywal) *Ny-1-*mediated HR to PVY where the virus was detected outside the cell death zone, that was able to reinitiate infection. This suggests that HR cell death is separated from the resistance mechanisms in this pathosystem [55]. Interestingly, a recent study showed that in a breeding clone carrying *Ny-1* duplex, smaller lesions were developed and a lower amount of virus was accumulated, as compared to plants of cv. Rywal (*Ny-1* simplex) after PVY^NTN^ inoculation, indicating the effect of gene dosage on the efficiency of PVY restriction [56].

One of the earliest hallmarks of HR is the rapid and intense production of reactive oxygen species (ROS) [57]. In *Ny-1*-mediated resistance of cv. Rywal, detection of large amounts of hydrogen peroxide and upregulation of several genes involved in redox state regulation from 1 day post-inoculation (dpi) indicated the importance of ROS signalling in an efficient HR ([58], Figure 2). Recent spatiotemporal analysis of responses in and surrounding the foci of viral infection, on ultrastructural, biochemical, and gene expression levels revealed that tight spatiotemporal regulation of redox state maintenance is required for the successful arrest of the virus [20]. Respiratory Burst Oxidase Homolog D (RBOHD) regulates spatial distribution of SA accumulation and is indispensable for viral arrest [20]. RBOHD protein accumulation and distribution were also different in HR-mediated resistance and sensitive response in cv. Sárpo Mira carrying *Ny-Smira* [59].

MAPK cascades are conserved signaling pathways across eukaryotes that mediate intracellular responses by transducing the extracellular stimuli downstream from the receptors. They are implicated in the signaling of plant developmental programes and response to multiple environmental stressors s and they are a core mediator for HR-associated PCD (reviewed by [60]). Specifically, the MAPK signaling network in HR-conferred resistance to PVY involves MAPK kinases 6 (MKK6) and the downstream targets MAPK4_2 (orthologue of AtMAPK4, -11 and -12), MAPK6 (orthologue of AtMAPK6), and MAPK13 (orthologue of AtMAPK13) [61]. Downregulation of *MKK6* was shown to increase PVY concentrations in infected plants, confirming that this kinase is an important component of potato immunity against the virus [62].

SA is a phenolic compound primarily recognized for its role in local defense induced against biotrophic and hemi-biotrophic pathogens. SA biosynthesis is triggered during PTI and ETI [63] and is required for the restriction of pathogens during HR in various pathosystems including viruses ([64], reviewed by [54]). SA was shown to be synthesized de novo in *Ny-1-*mediated resistance to PVY^N-Wilga^ in potato cv. Rywal [58], (Figure 2). The lack of SA accumulation led to unrestricted viral spread accompanied by fast lesion expansion [55]. SA was shown to orchestrate molecular events leading to resistance on the transcriptional level, where the delayed onset of defense responses and perturbation of hormonal signalling were of most importance. SA biosynthesis genes were strongly upregulated in the nontransgenic plants, correlated with SA content increase. Moreover, brassinosteroid and gibberellin (GA) biosynthesis were upregulated only in the early stages of efficient HR response [58]. On the other hand, in SA-deficient genotype, expressing salicylate hydroxylase (NahG), NahG-Rywal, the SA biosynthesis induction responded in narrower timespan and JA and ethylene biosynthesis were induced faster compared to nontransgenic genotype [58]. SA was also shown to regulate the transcriptional events in and around the foci of viral amplification as, for some genes, spatiotemporal regulation was lost or, for some, altered in SA-depleted plants [20].

The successful viral arrest could be linked to responses in primary metabolism, albeit for this aspect of response extensive mechanistic insights are not yet available. Early induction of photosynthesis-related genes, that could be related to increased energy demands for efficient defense response, was observed in HR of potato with PVY [58,65] (Figure 2). Similarly, granule bound starch synthase gene was induced only in the tissue surrounding viral foci in incompatible interaction [20].

The mechanism that actually blocks viral spread remains elusive in HR against PVY. Cell wall reinforcement is a frequent plant response to viral infection to act as a physical barrier and is thought to block or delay local and systemic movements of viruses [66]. Genes involved with cell wall rearrangement were upregulated 1 dpi in *Ny-1-*mediated HR resistance response but not in SA-deficient NahG plants [58]. On the contrary, *xyloglucan endotransglucosylase-hydroxylase XTH9* was downregulated in the first hours in HR-mediated resistance response of cv. Premier Russet to PVY^O^ [65]. In the *Ny-Smira-*mediated HR resistance response to PVY^NTN^, localization and abundance of beta-glucosidase, extensin, cellulose synthase, and xyloglucan xyloglucosyl-transferase 10 dpi was associated with cell wall strengthening [67,68] (Figure 2).

In Arabidopsis, constitutive, but not stress-related, β-1,3-glucanases were implicated in enhancing virus spread by degrading callose at plasmodesmata [45]. Interestingly, similar function, but of PVY-induced *β-1,3-glucanase from class III* (*Glu-III*), was shown in potato in different cultivars [69]. Moreover, callose deposition that restricted both local and systemic spread of PVY^O^ was triggered by the helper component proteinase (HC-Pro) in a temperature-dependent manner in HR response of cv. Russet Burbank, carrying the so far uncharacterized *Ny* gene [70] (Figure 2). Although callose deposition appeared to demarcate the lesion cells from the surrounding tissue from 3 dpi with PVY^N-Wilga^ [58], the virus was found outside of the formed callose ring, indicating that callose might not be blocking the spread of the virus to the adjacent necrotic lesions in *Ny-1-*conferred HR [55].

Specific for the HR response of cv. Premier Russet [65] and cv. Rywal [58], was also an early induction of cysteine proteinase inhibitors that could be related to the inhibition of virus multiplication [71].

### 2.2. Extreme Resistance Response: No Signs of Battle

In the case of ER, potato plants show no symptoms (Figure 1) or very limited necrosis (in the form of pinpoint lesions) in some genotypes [72]. Viral amplification cannot be detected by standard molecular methods due to either inhibition of virus multiplication in the infected cells or restricted cell-to-cell movement of the virus [13,73].

To date, several resistance genes conferring ER to PVY were identified and mapped to potato chromosomes IX, XI, and XII. They were introduced into potato cultivars from wild or domesticated *Solanum* species. Two alleles (*Ry_sto_* and *Ry-f_sto_*) [74,75] are derived from *S. stoloniferum, Ry*_ch_*_c_* gene from *S. chacoense* [76] and *Ry**_adg_* from *S. tuberosum ssp. Andigena* [77]. The *Ry_hou_* gene was found in *S. hougasii* [78] but it has not been used for breeding. Additionally, the *Ny-DG* gene present in the diploid potato clone DG-68 features ER-like response [79]. Recently, a dominant resistant gene *Ry(o)_phu_* was identified in *S. tuberosum* ssp. Phureja, conferring resistance to PVY^o^, PVY^NTN^, and PVY^N-Wi^ strains [80]. The introduction of R genes into cultivated potato has shown to confer durable resistance against several PVY strains [40,81]; however, the number of potato cultivars carrying *Ry-genes* is relatively low (see review [40]). Although the phenotypic outcome is the same in all reported cases of extreme resistance (Figure 1), the underlying mechanisms might differ. In genotypes carrying *Ry(o)_phu_* and *Ry_sto_* genes amplification of PVY^N605^ (infectious clones carrying GUS) could be detected in limited areas, while in cv. Tacna (carrying *Ry_adg_-*gene) no amplification of the virus was detected [80].

Only recently, the first R gene against PVY in potato, the *Ry_sto_* gene, conferring ER response to PVY was isolated and functionally characterized [82], opening new opportunities for improved breeding strategies. The gene encodes for an intracellular nucleotide-binding leucine-rich repeat (NLR) receptor with an N-terminal Toll/interleukin-1 receptor (TIR) domain (TIR-NLR) and mediates immunity in potato and tobacco plants against different strains of PVY and potato virus A. Sequence alignment of *Ry_sto_* and *Ry-f_sto_* alleles showed 100% homology between them. According to the authors, the elicitor of *Ry_sto_-*mediated ER is PVY CP. They expressed the viral proteins in transgenic *Ry_sto_* and wild type tobacco plants and only CP induced strong HR [82]. These results are in disagreement with previously published studies. Using a similar approach, Mestre et al. showed that NIa is required for elicitation of *Ry_sto_*-mediated HR in potato [83,84]. Further studies are required to determine the nature of this discrepancy in reported mechanisms.

Recently, another R gene against PVY was isolated from pepper (*Capsicum annuum)* [85]. The authors cloned and characterized the dominant gene *Pvr4,* which confers ER to a broad range of potyviruses, including PVY [86]. It encodes a protein belonging to the LRR receptor class with an N-terminal coiled-coil domain (CC-LRR). It was previously reported that NIb of pepper mottle virus (PepMoV) is the avirulence factor inducing HR in *Pvr4-*mediated resistance [87]. In the recent study, the interaction between Pvr4 and NIb from PepMoV was confirmed and the candidate gene was validated in *N. benthamiana* [85].

The isolation of *Ry* genes also shed some light on the downstream signalling pathways involved in ER. Most TIR-NLRs activate resistance in a lipase-like protein Enhanced Disease Susceptibility 1 (EDS1)-dependent manner [88,89]. Recently, it was reported that, in some cases, also the CC-NLR protein N requirement gene 1 (NRG1) is required to activate immunity [90,91]. Recently, Grech-Baran et al. demonstrated that the Ry_sto_-mediated resistance depends on both, EDS1 and NRG1 [82]. They additionally showed that SA signalling is not required for an efficient ER, differentiating this ER interaction from a typical HR (Figure 2). Unusual for TIR-NLRs, the resistance is not temperature-sensitive [82].

In the soybean–soybean mosaic virus pathosystem, a recent study suggested that the primary mechanism of ER against this potyvirus is the inhibition of viral cell-to-cell movement by callose deposition in an abscisic acid signalling–dependent manner [73]. However, the information on mechanisms of ER response against PVY is limited. A comparative transcriptomics study of the early response to PVY infection in a sensitive cv. Igor and a Ry_sto_ bearing resistant cultivar cv. Santé showed that cv. Santé had a higher number of differentially expressed genes at at 0.5 h post-infection (hpi), while cv. Igor exhibited the greatest response at 12 hpi. The faster perception and signaling in the ER response might enable a more efficient defense reaction [92]. In fact, only the resistant cultivar exhibited a very early deployment of alkaloidal defencses and pathogenesis-related (PR) proteins. The induction of brassinosteroids synthesis, lignin, and polyamine pathways and proteinase inhibitors were also among the features specific for ER in cv. Santé to PVY infection [92]. However, these observations are based on changes in transcript abundance. Translation between transcriptomics and proteomics data is not trivial, as transcriptomic data may not necessarily reflect protein abundance in these functional categories [93]. Recently, a proteomic analysis of potato ER response against PVY was published by Szajko et al. where the authors showed that the stress-responsive proteins were the most abundant among the qualitative changes induced by PVY in the resistant genotype PW363 [56].

The role of *Glu-III*, a gene that was found to be induced in response to PVY infection [18,92,94], was selected for further functional analysis. The possibility of breaking the resistance and facilitating the viral spread by *Glu-III* over-expression was studied in the cv. Santé. The results indicated that a transient multiplication of virus could occur in callose deficient plants; however, later the virus was blocked by Ry_sto_ gene signalling [69] (Figure 2).

### 2.3. S-Gene Conferred Resistance: Can’t Live without You

Analysis of naturally occurring recessive resistance against potyviruses in crop plants has revealed that often the genes behind the resistance phenotype are either translation initiation factor *eIF4E* or *eIFiso4E* gene (Figure 2). Naturally existing resistant versions of eIF4E differ from the susceptible form of the protein by only one to five amino acid changes, near the VPg-binding region (reviewed by [95]). The tobacco isoform *eIF4E-2* expression level is positively correlated with resistance durability and might act as a decoy, limiting the ability of PVY to evolve towards resistance breaking [96]. Nevertheless, resistance-breaking strains of PVY, carrying a mutation in the VPg, that overcome this type of resistance, were identified. Takakura et al. showed that eIF(iso)4E-T isoform is required for infection by the resistance-breaking strain of PVY in tobacco [97].

To date, no eIF4E-related, naturally occurring resistance, was found in potato. Although it may exist, it would be difficult to uncover and maintain in breeding programs due to potato polyploidy (reviewed by [40]). Therefore, in potato, this type of resistance to PVY^O^, PVY^N^ and PVY^NTN^, was established by heterologous expression of pepper *eIF4E* or engineering the existing *eIF4E* gene [98,99]. Transcriptional analysis of cv. Atlantic constitutively expressing a modified *eIF4E* showed that this over-expression suppressed the endogenous *eIF4E* allele but also deregulated the expression of genes involved in oxidative species homeostasis and stress responses [99].

The function of S-genes is either to support viral infection cycle or to be negative regulators of immunity in plants (Figure 2). A S-gene, with the function in immune response, was recently identified in the interaction of PVY^N^ with tobacco [100]. Parallel analysis of messenger RNA (mRNA) and small RNA (sRNA) profiles discovered several PVY-derived small interfering RNAs which target the host gene *tobacco translationally controlled tumor protein* (*TCTP*) involved in stabilization of the ethylene receptor. The silencing of *TCTP* suppressed the PVY infection, whereas the over-expression of *TCTP* increased plant susceptibility confirming its role as a susceptibility factor [100].

## 3. Susceptible Response to PVY

In a susceptible interaction, the plant innate immune system is not able to establish an efficient defense response and the virus can replicate and invade the plant [12]. Upon entry into a susceptible cell, PVY initiates the replication cycle that consists of a chain of several partially overlapping events including translation, replication, cell-to-cell movement, antiviral defense/counterdefense, and encapsidation. The complex regulatory network that must be established between potyviral and host proteins to ensure a successful infection was reviewed before [101]. Afterwards, the virus moves into neighboring cells, leaf veins, and vascular system. Once into the vasculature, the virus infects distant parts of the plant [102]. The consequence of the initial infection and the spread of the virus is the appearance of symptoms in case of sensitive genotypes. Thus, sensitivity is often associated with important potato crop losses in terms of yield and quality. In contrast, a tolerant host is a plant that the virus can infect (replicate and spread) without causing severe symptoms [12]. In both cases, the infection by PVY causes a vast reprogramming of the host cell that results in cytological, biochemical, and physiological changes.

### 3.1. Sensitive Response: Too Late or too Weak

In contrast to plant–virus interactions, in sensitive interactions, virus infection does not trigger efficient and/or timely resistance response. Systemic disease symptoms manifest later in the infection in the upper noninoculated leaves and usually do not impede virus multiplication or its systemic movement. The disease symptoms (Figure 1) are a consequence of failing HR-like programmed cell death response, cellular rearrangements caused by viral multiplication such as cell death resulting from unfolded protein response [103] or redistribution of resources due to both viral multiplication, and unsuccessful runaway immune response [104]. Consequently, the molecular events leading to similar disease phenotype can be different [65,105].

PVY is recognized by the plant by receptor proteins also in susceptible interaction [14], triggering the immune response, but the response is not able to restrict the virus (Figure 2). Several studies comparing sensitive and resistant interaction showed the importance of timing for efficient defense response. From 10 hpi, the transcriptomic responses in resistant and sensitive interactions differentiated and featured strong but slower activation of defense-related genes in the sensitive interaction [58,65,92,105]. For example, in SA-deficient NahG-Désirée and cv. Igor, the expression of *PR* genes was even stronger than in symptomless cv. Désirée and the dynamics of response was slower, compared to the resistance response, peaking two days later [58,94]. The dynamics of changes can differ also if the same genotype is infected with different viral strains. ROS signalling was induced in interactions of potato cv. Igor with PVY^NTN^ and PVY^N^, and in *N. benthamina* interaction with PVY^O^. However, the mild isolate PVY^N^ induced a more rapid response [106] or a very weak one in case of PVY^O^ [22]. Additionally, in interaction of PVY^O^ with tomato or *N. benthamiana* (in both pathosystems mild symptoms develop), late response of ethylene, JA, and ROS signaling was detected, while induction of *PR-1* was very weak [22,24].

An important feature of sensitive interactions is a response to the hijacked metabolism. The transcriptional analysis of responses to PVY^NTN^ in secondary infected tubers of a sensitive cv. Igor was strongly dependent on virus concentration. Viral concentration was the highest in the necrotic lesions, where a massive transcriptional response was observed. Genes for proteases with caspase-like activity that are markers of cell death were induced, as were also endoplasmic reticulum chaperones, markers of the endoplasmic reticulum stress response [105] (Figure 2), which is a known consequence of intensive viral multiplication [101]. Similarly, in the symptomatic part of tobacco leaves infected with PVY, the concentrations of ROS and *PR* gene induction was the strongest and correlated with detected high concentrations of the virus [14].

A consequence of resource allocation in the sensitive interaction is often manifested as downregulation of genes involved in photosynthesis and chlorophyll biosynthesis [18,22,58,92,107] that was also linked with the decrease of photochemical efficiency from the onset of PVY multiplication in the sensitive NahG-Désirée [107] (Figure 1). The collapse of photosynthesis leads to the metabolic transition from source to sink, manifested as activation of pentose phosphate pathway, lipid β- oxidation, and amino acid mobilization in SA-deficient NahG Rywal 6 dpi with PVY^N-Wilga^ [22,58]. Similarly, accumulation of sugars and phenylpropanoids and alterations in the Krebs cycle and γ-aminobutyric acid-shunt activities were found to be very pronounced at 6 dpi in the compatible interaction of cv. Igor with PVY^NTN^ and PVY^N^, corresponding to the time of the strong viral multiplication [106].

The observed responses can also be the consequence of viral counterdefense. For example, it was shown that the HC-Pro of the necrotic PVY strain is able to suppress callose deposition in cv. Premier Russet [70].

### 3.2. Tolerant Response: Balance between Plant Defense and Virus Counterdefense

Tolerance is defined as an interaction in which viruses accumulate to some degree without causing significant loss of vigor or fitness to their hosts. It can be described as a stable equilibrium between the virus and its host, an interaction in which each partner not only accommodates trade-offs for survival but potentially also receive some benefits (reviewed by [108]).

On the molecular level, tolerant responses to PVY included attenuated expression of defense response genes (such as *PR-1*, *PR-2*, Figure 2) [14,107,109]. In the interaction of tolerant cv. Désirée with PVY^NTN^, photosynthesis genes were shown to be transiently induced at early stages of infection but then rapidly repressed at the onset of virus multiplication [107]. It was suggested that the early induction of photosynthesis (and other primary metabolism-associated genes) helps to promote the induction of tolerant responses (Figure 2). In the same pathosystem, sRNAs have been shown as an important regulatory level in the establishment of tolerance [110], leveling both, responses in immune signaling and GA regulatory network (Figure 2). Decreased levels of miR6022 were linked with upregulation of its predicted target genes encoding LRR-RLKs, which were shown to be regulated similarly in some mutualistic symbiotic interaction [111]. In addition, tolerant plants exhibited increased levels of sRNAs targeting transcripts encoding two GA biosynthesis genes as well as of the transcript encoding MYB33, an orthologue of gibberellin- and abscisic acid-regulated MYB (GAMYB) transcription factor involved in GA signal transduction. The level of GA was decreased, while the levels of other hormones were not affected. This indicates that, in tolerant response, repression of GA signaling is coupled with an increase in immune receptor gene expression which was corroborated by a discovery of GAMYB binding sites in the *MIR6022* promoter region [110].

The dynamic regulation of miRNAs and secondary sRNAs was also observed in late responses of tomato to PVY^C^, which suggested a functional role of sRNA-mediated defenses in the recovery phenotype [112], which is an inducible form of tolerance.

Although one characteristic of tolerant interaction of potato with PVY is the lack of SA induction after virus infection [109,113], SA still has an important role in the establishment of tolerance (Figure 2). This role was manifested by transgenic NahG-Désirée plants, that showed severe symptoms upon virus infection, stronger *PR* gene expression [94], and a diminished induction of photosynthesis genes at early stages of infection [107]. Moreover, sRNA-regulated GA-R gene link is abolished in this genotype [110].

The difference between tolerant and sensitive response can also lie in recognition of the virus by receptor proteins. In tobacco cv. Samsun NN interaction with PVY^N^ systemic veinal necrosis appear. A single mutation in the *TPN1* R gene abolishes the systemic veinal necrosis symptoms and also reduces the detected immune responses [14]. Similar lack of recognition, leading to tolerance, was observed also in studies of Hop/Sti1 [109], a multifunctional cochaperone that has been implicated in the maturation of RLK involved in PTI sensing of chitin. Silencing of this cochaperone prevents viral recognition and, consequently, attenuated induction of SA accumulation, ROS production and transcriptional activation of *PR* genes was observed in the symptomless plants.

Tolerance to viruses has been mapped to a single or multiple genes (reviewed by [108]). Some are most probably inefficient receptor proteins (as described by [14]), while others might be linked to the establishment of tolerance due to changes in immune signaling [109,110] or balance between primary metabolism and defense [110].

## 4. Future Outlook

PVY is the most damaging virus of potato and other Solanaceae crops worldwide. Thus, understanding the molecular mechanisms governing the outcome of the interaction is essential for sustainable production in a changing environment. During the last decade, important technical improvements have slowly improved our knowledge on general mechanisms of defense response in PVY infection. This should improve even further with the implementation of mathematical modelling to predict responses of immune signaling [28] and linked primary metabolism. In addition, the production of PVY infectious clones [114,115,116] and advances in genetically encoded sensors in plants [117] will enable spatiotemporal monitoring of interactions between host and viral proteins during viral infection. The advanced genome editing technologies will allow an easier functional analysis of generated hypotheses [118].

With this toolkit in hand, we will be able to fill in the knowledge gaps and provide answers for remaining questions. For example, it is still not understood how PVY multiplication and/or cell-to-cell movement are blocked in resistant interaction. We know it is not the PCD that is blocking the viral movement [55], as was already shown in other viral pathosystems [54]. The only so far discovered mechanism blocking the virus spread is deposition of callose in plasmodesmata [45,119] that was not yet confirmed or rejected for potyviruses. However, in some cases of resistance (e.g., cv. Tacna carrying *Ry_adg_* [80]) viral multiplication is not detected, indicating that different mechanisms blocking viral infection might be in place in different pathosystems. It is clear that the mechanistic basis of resistance signaling differs depending on the R-protein involved (e.g., SA-dependent or SA independent). So, perhaps in the future, classification of pathosystems according to the signaling network triggered might be more informative than current phenotypic classification to HR and ER resistance. What we also know is that delicate spatiotemporal regulation is required for the efficient response [20]. So, perhaps a balance of several mechanisms is required to block the virus.

Another question remaining is whether, in plant, PVY is perceived by multiple receptors. This was indicated in resistant interaction governed by *Ry(o)_phu_* [80], where extensive analysis of several biparental cross-populations was performed. Perception of PVY was also confirmed to occur in susceptible interactions [14,109]. These data, albeit still scarce, make it plausible that perception involving multiple receptors is present on a broader scale. This notion would, however, change our way of understanding the immune signaling network triggering.

Related to that, how the background genotype (e.g., characteristics of the network responding downstream of the receptor proteins) affects the efficiency of resistance it is also widely unstudied. It is now generally assumed that, if the correct R-gene is incorporated into the genetic background, we will obtain the desired phenotype. The importance of the availability of certain components of downstream immune signaling was confirmed for a cochaperone involved in the maturation of RLKs [109] and for the accumulation of SA [58,94]. The difference in these components can switch the interaction modes between resistant and susceptible or tolerant and sensitive.

The next important question is: What are the mechanisms involved in the establishment of tolerance? Resistance is traditionally preferred over tolerance in breeding programs, as tolerant crops represent a virus reservoir. Nevertheless, tolerance may have an advantage over resistance for crop protection because it does not actively prevent virus infection and/or replication, therefore there is little evolutionary pressure for the virus to mutate and to evolve into more aggressive strains ([120], reviewed in [108]). It was already determined in one PVY pathosystem that the disease is the consequence of a runaway immune response, which can be easily blocked by mutation of receptor protein [14]. This, however, is not true for all pathosystems [107] and more data providing a holistic understanding of the underlying processes are required. Even more powerful and with a higher agricultural value, would be the identification of S-genes (reviewed in [95]). For both approaches, a detailed understanding of the balance between growth and immunity would be required as quite often the introduction of S-genes mediated resistance or tolerance comes with the penalty of growth and consequently, has an impact on crop harvest.

Through the design of experiments aimed to answer these questions, we will provide knowledge that will allow for the development of efficient breeding strategies as well as for improvement of agricultural management strategies.

## Figures and Tables

**Figure 1 viruses-12-00217-f001:**
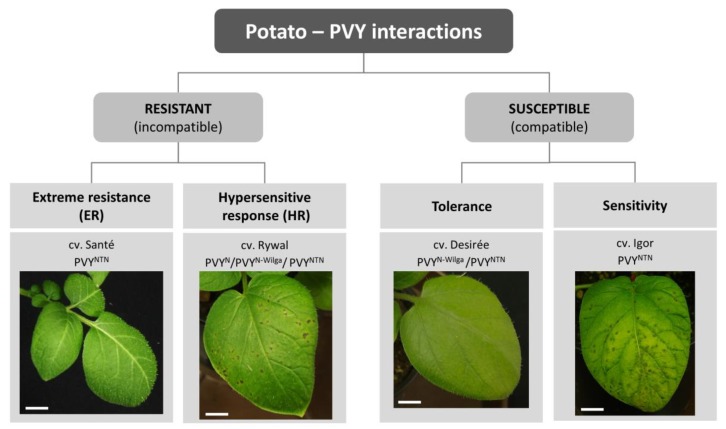
Outcomes of potato-potato virus Y (PVY) interaction. Outcomes depend on the host genotype, viral strain, and environmental conditions, and are manifested as different responses in terms of virus multiplication and disease symptoms’ development. Photos present symptoms appearing on the inoculated leaves 6 days after inoculation in selected potato cultivars in optimal environmental conditions. Examples of viral strains that give the same outcome are shown for each cultivar. Scale = 1 cm.

**Figure 2 viruses-12-00217-f002:**
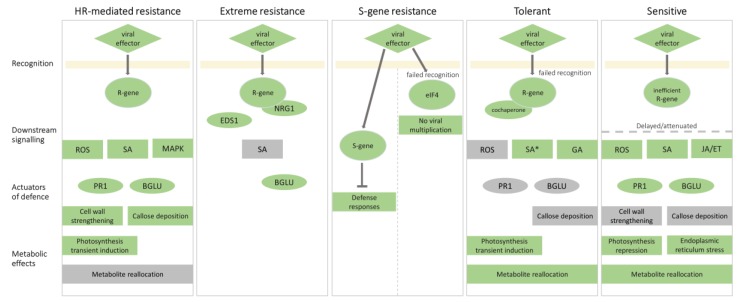
Comparison of molecular responses to PVY resulting in different outcomes. Molecular events involved in virus recognition, downstream signaling, defense responses, and metabolic effects in resistant and susceptible outcomes are presented. The components (genes, proteins, and processes) that were shown to be involved in a particular outcome are highlighted in green, while the components that were shown not to be involved in a specific outcome are highlighted in grey. In hypersensitive response (HR)-mediated resistance, the viral effector (e.g., helper component proteinase, HC-Pro) is recognized by the R-gene (e.g., *Ny-1* or *Ny-Smira*), which activates reactive oxygen species (ROS) signaling (mediated by, e.g., Respiratory Burst Oxidase Homolog D, RBOHD), mitogen-activated protein kinase kinase (MAPK) cascades (e.g., MAPK4_2, MAPK6, MAPK13, and MAPK kinase 6, MKK6) and salicylic acid (SA) signaling. Resistance is mediated by changes in *PR1* (*pathogenesis-related gene 1*) and *BGLU* (*β-1,3-glucanase gene*) expression, cell wall strengthening, and callose deposition. Transient induction of photosynthesis is observed. In extreme resistance, the viral effector (e.g., HC-Pro) is recognized by the R-gene (e.g., *Ry_sto_*), but N requirement gene 1 (NRG1) is required for activation of downstream signalling, which depends on Enhanced Disease Susceptibility 1 (EDS1). *BGLU* (*Glu-III*) is involved in viral restriction. S (susceptibility)-gene mediated resistance depends on either a mutation in eukaryotic translation initiation factor 4 (eIF4), which prevents viral multiplication, or on an altered S-gene, which inhibits defense responses. Tolerant interaction can be a consequence of failed recognition of an R-gene (e.g., *TPN1*) or cochaperone (e.g., Hop/Sti1) required for virus multiplication. Downstream signaling involves gibberellins (GA). SA is not increased but has an important role in the establishment of tolerance. ROS, *PR1, BGLU,* and callose deposition are not involved. Photosynthesis transient induction and metabolite reallocation are observed. In sensitive interaction, the viral effector is recognized by an inefficient R gene, leading to delayed or attenuated downstream ROS, SA and jasmonic acid/ethylene (JA/ET) signaling, and *PR1* and *BGLU* induction. Cell wall strengthening and callose deposition are absent. Metabolic effect feature photosynthesis repression, endoplasmic reticulum stress and metabolite reallocation. Other mechanisms, leading to this outcome are possible.

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
