# Peer review of "Plant Molecular Responses to Potato Virus Y: A Continuum of Outcomes from Sensitivity and Tolerance to Resistance"

_viruses, 2020, doi:10.3390/v12020217_

Round 1

Reviewer 1 Report

Manuscript viruses-721494 reviews the literature on interactions between potato virus Y (PVY) and potato in terms of phenotypic outcomes that range from susceptibility to resistance.  The article is of interest although editorial changes should be considered for improvement.  See recommendations below:

Line 3: Change sensitivity to susceptibility

Line 11: Eliminate infection

Line 144: Add tolerence to the list of specific outcomes

Line 18: Change Virus to virus

Line 22: Eliminate plant

Line 23: Change harvest to production

Line 24: Eliminate Moreover

Line 25: f... from the Solanaceae family ...

Line 28: ... of the genus Potyvirus, family Potyviridae ...

Line 38: ... the viral structural protein is involved in several steps of the virus infection cycle, including ...

Line 40: Change stablishing to establishing

Lines 43-44: ... efficient infection, of mechanisms of plant defence and viral counter defence, to circumvent ...

Line 45: ... mechanisms defines a specific ...

Line 51: Eliminate described

Line 53: In potato, ER ...

Line 94: ... responses to PVY resulting in different ...

Line 312: Takakura et al. showed ...

Line 439: Eliminate [116-117]

Line 441: ...viral proteins during infection.  The advanced ...

Lines 445-446: We know the programmed cell death blocks viral movement ...

Line 447: Eliminate either

Line 448: Change Potyviruses to potyviruses

Line 456: ... is whether PVY is perceived by multiple receptors in plants

Line 462-483: related to that, how teh background genotypes (e.g., characteristics of network responding downstream receptor proteins) affects the efficiency of resistance is also widely unstudied.

Line 483:  ... efficient breeding strategies, as well as improvement ...

Author Response

Manuscript viruses-721494 - response to Reviewer 1

We are grateful for the reviewer's comments that will contribute to the quality of the manuscript. Most of the proposed changes were on point and were therefore implemented in the revised manuscript. However, in some cases, the proposed change would alter our intended meaning. As misunderstanding was obviously a sign of ambiguity, we have reformulated the sentence/phrases in question. We are responding to the reviewer’s comments point-by-point below (the comment in italics, our response in regular text). Besides the proposed changes, we have made some other needed changes (minor grammar and gene naming fixes as well as fixing a mistake in Figure 2). All the changes are marked with “tracked changes” in the revised manuscript.

Manuscript viruses-721494 reviews the literature on interactions between potato virus Y (PVY) and potato in terms of phenotypic outcomes that range from susceptibility to resistance.  The article is of interest although editorial changes should be considered for improvement.  See recommendations below

Line 3: Change sensitivity to susceptibility 

Line 11: Eliminate infection

corrected

Line 144: Add tolerence to the list of specific outcomes

We believe that here, line 14 and not 144 was meant. As outlined in the previous response, the two main responses are resistance and susceptibility, and sensitivity and tolerance are its hyponyms. Nevertheless, using different words in the title and abstract might be confusing for the reader, therefore we changed the text in line14 from  »…that can be resistance or susceptibility« to »..that can be resistance, sensitivity or tolerance« to encompass all the possible outcomes.

Line 18: Change Virus to virus

corrected

Line 22: Eliminate plant

corrected

Line 23: Change harvest to production

corrected

Line 24: Eliminate Moreover

corrected

Line 25: f... from the Solanaceae family ...

corrected

Line 28: ... of the genus Potyvirus, family Potyviridae ...

corrected

Line 38: ... the viral structural protein is involved in several steps of the virus infection cycle, including ..

corrected

Line 40: Change stablishing to establishing

corrected

Lines 43-44: ... efficient infection, of mechanisms of plant defence and viral counter defence, to circumvent ...

We believe that the proposed change would change our meaning. Nevertheless, we have corrected the sentence to be more clear and it now reads: »The complex interaction between plants and viruses includes the interplay of several mechanisms: 1) viral hijacking of host factors required for efficient infection, 2) mechanisms of plant defence and 3) mechanisms of viral counter defence, to circumvent plant defence.«

Line 45: ... mechanisms defines a specific ...

corrected

Line 51: Eliminate described

corrected

Line 53: In potato, ER ...

corrected

Line 94: ... responses to PVY resulting in different ...

corrected

Line 312: Takakura et al. showed ...

corrected

Line 439: Eliminate [116-117]

corrected

Line 441: ...viral proteins during infection.  The advanced ...

corrected

Lines 445-446: We know the programmed cell death blocks viral movement ...

The proposed change would change our intended meaning that PCD is actually not stopping PVY. The question if the programmed cell death (PCD) of a limited region of cells restricts pathogen proliferation or if lesion formation is merely collateral damage of the resistance mechanisms has long been discussed and it seems that that the role of cell death in resistance depends on the type of host-pathogen interaction (see review by Künstler, Physiol. Mol. Plant Pathol. 2016). In plant-virus pathosystems, cell death was shown to be uncoupled from resistance in the HR in different host-virus interactions involving PlAMV, PVX, CaMV, TBSV, CMV, TMV and ToMV (Chu et al., 2000; Chandra- Shekara et al., 2004; Cawly et al., 2005; Ishibashi et al., 2007; Komatsu et al., 2010; Liu et al., 2010; Hafez et al., 2012; Ando et al., 2014). Our recent study (Lukan et al., Frontier Plant Sci, 2018) confirmed that PCD is not stopping the virus in Ny-1 mediated HR against PVY, where the virus was detected outside the cell death zone, that was able to reinitiate infection. These findings are presented in the HR section of the manuscript (lines 181-182). Therefore, we would not like to change this statement but have, however, amended the sentence with information about the other viral pathosystems. The corrected sentence now reads:

“We know it is not the PCD that is blocking the viral movement [55], as was already shown in other viral pathosystems [54].”

Line 447: Eliminate either

corrected

Line 448: Change Potyviruses to potyviruses

corrected

Line 456: ... is whether PVY is perceived by multiple receptors in plants

To emphasise the fact that several receptors recognizing PVY could exist within the same plant we haven't accepted your change but have rather changed the sentence to improve clarity. It now reads: »Another question remaining is whether in plant PVY is perceived by multiple receptors«.

Line 462-483: related to that, how teh background genotypes (e.g., characteristics of network responding downstream receptor proteins) affects the efficiency of resistance is also widely unstudied.

corrected

Reviewer 2 Report

PVY is one of the most important viruses infecting Solanaceae plants. In this review manuscript, authors focus on plant responses against PVY infection during HR, ER, or susceptible infection. Topics include recent progresses and provide a broad knowledge in the field. I have several comments listed below for revision.

Title: Specify subject of “molecular responses”, for example, plants, potato varieties, etc.

Figure 2: No difference is shown between "HR" and "Sensitive" for viral effector and R-gene interactions. Please modify to emphasize the difference.

Line 94; Correct “PV” to “PVY”.

Line 129: “fully understood” should be changed to “well understood” because what is “full” would never be unknown.

Section 2.1: Many genes are enumerated independently. A paragraph summarizing a comprehensive view of the section would help non-specialist readers.

Section 2.2: Please describe what is the differences of molecular responses between HR and ER.

Line 340: cytological, biochemical, and physiological changes are not necessarily result from transcriptome reprogramming. For example, lipid metabolisms are changed by actions of viral proteins.

Author Response

Manuscript viruses-721494 - response to Reviewer 2

We are grateful for the reviewer's comments and suggestions that will contribute to the quality of the manuscript. We were able to implement most of the proposed changes and take all the comments into consideration. We are responding to the reviewer’s comments point-by-point below (the comment in italics, our response in regular text). Besides the proposed changes, we have made some other needed changes (minor grammar and gene naming fixes as well as fixing a mistake in Figure 2). All the changes are marked with “tracked changes” in the revised manuscript.

PVY is one of the most important viruses infecting Solanaceae plants. In this review manuscript, authors focus on plant responses against PVY infection during HR, ER, or susceptible infection. Topics include recent progresses and provide a broad knowledge in the field. I have several comments listed below for revision.

Title: Specify subject of “molecular responses”, for example, plants, potato varieties, etc.

Figure 2: No difference is shown between "HR" and "Sensitive" for viral effector and R-gene interactions. Please modify to emphasize the difference.

We agree that a distinction in the mechanism should be visible on the image. Therefore, we have modified Figure 2 to distinguish an efficient recognition and signalling that is happening in the HR from the inefficient recognition and delayed or attenuated signalling that occur in the sensitive outcome. As of note, in Figure 2, only one possible mechanism, involved in the sensitive interaction, is shown, although there can be several, as pointed in section 3.1 (lines 368) and discussed in section 4 (lines 480-87).

Line 94; Correct “PV” to “PVY”.

corrected

Line 129: “fully understood” should be changed to “well understood” because what is “full” would never be unknown.

corrected

Section 2.1: Many genes are enumerated independently. A paragraph summarizing a comprehensive view of the section would help non-specialist readers.

We agree that the situation is complex and might be difficult to understand for a non-specialist reader. Although we have tried to explain the roles of the individual genes in the text, we have now added the summary of the described mechanisms in the individual outcomes and included all the mentioned genes to the legend of Figure 2 for better clarity.

Section 2.2: Please describe what is the differences of molecular responses between HR and ER.

The mechanisms of HR and ER are difficult to compare as the studies of mechanisms involved in ER are limited. The most straightforward difference is the independence of SA-signalling in Rysto mediated ER that was recently reported and is shown in Figure 2. To emphasise this, we have amended a sentence in line 295 that now reads: “They additionally showed that SA signalling is not required for an efficient ER differentiating this ER interaction from a typical HR (Figure 2)”. Moreover, the characteristics can be compared in Figure 2. Nevertheless, as suggested by Torrance et al, (2020) an ER phenotype could be the result of different downstream signalling events, leading to very little in some cases or no viral multiplication. We address this issue in the “Future outlook” section (lines 468-472), suggesting that “classification of pathosystems according to signalling network triggered might be more informative than current phenotypic classification to HR and ER resistance”.

Line 340: cytological, biochemical, and physiological changes are not necessarily result from transcriptome reprogramming. For example, lipid metabolisms are changed by actions of viral proteins.

We agree that “transcriptome” was too specific in this sentence, therefore, we have changed the sentence that now reads: “In both cases, the infection by PVY causes a vast reprogramming of the host cell that results in cytological, biochemical and physiological changes”.